# A Roadmap for the Certification of Polyurethane Flexible Connectors Used as Envelope Products in the Next Generation of Healthy, Nearly Zero-Energy Buildings

**DOI:** 10.3390/ma17225503

**Published:** 2024-11-12

**Authors:** Aneta Nowak-Michta, Arkadiusz Kwiecień, Jagoda Michta

**Affiliations:** 1Faculty of Civil Engineering, Cracow University of Technology, 24 Warszawska St., 31-155 Cracow, Poland; 2FlexAndRobust Systems Ltd., 24 Warszawska St., 31-155 Cracow, Poland; ak@flexandrobust.com; 3Faculty of Environmental and Energy Engineering, Cracow University of Technology, 24 Warszawska St., 31-155 Cracow, Poland; jagoda.michta@student.pk.edu.pl

**Keywords:** polyurethane flexible connectors, certification path, PM&VL7 test line, CE marking, nearly zero-energy buildings

## Abstract

Currently, in the European Economic Area (EEA), producers of building materials are implementing innovative solutions that provide a chance for the widespread construction of zero-emission and zero-energy buildings. However, they encounter legal barriers related to the lack of standardization procedures enabling the rapid placement of innovative construction products on the market. The European project Horizon 2020: Measuring Envelope Products and Systems Contributing to the Next Generation of Healthy, Nearly Zero-Energy Buildings (MEZeroE) aims to support producers of innovative envelope products used in zero-energy buildings, including in the field of certification, CE (European Conformity), marking and placing them relatively quickly on the market. This article presents one of the research procedures developed and tested by Pilot Measurement & Verification Lines (PM&VL7) as part of the MEZeroE project for Flex&Robust polyurethane flexible connectors. This procedure considers the applicable legal requirements regarding CE marking and also indicates a certification path for this type of product.

## 1. Introduction

The Intergovernmental Panel on Climate Change (IPCC) predicts that if CO_2_ and other greenhouse gas emissions are not significantly reduced soon, global temperatures will increase by more than 1.5–2 °C this century [1]. Reducing greenhouse gas (GHG) emissions is key to combating global warming. In 2023, the total CO_2_ emissions from energy combustion and industrial processes amounted to 37.4 Gt [2]. In Europe, buildings are responsible for about 36% of energy-related greenhouse gas (GHG) emissions and 40% of final energy consumption [3].

One of the European Union’s (EU’s) actions in this area is the Green Deal, which aims to reduce greenhouse gas emissions in the EU by 55% by 2030 and achieve carbon neutrality by 2050 [4]. Achieving such ambitious goals requires significant changes in the construction sector [5]. The REPowerEU plan has set a decarbonization path and raised the renewable energy target to 45% [6]. Plans to achieve climate neutrality in Europe by 2050 are included in the revised version of the Energy Performance of Buildings Directive (EPBD) [7] and in the Energy Efficiency Directive (EED) [8,9]. The European Commission proposes to change the nearly zero-energy requirements into zero-emission requirements for all new buildings from 1 January 2030 and for all new buildings occupied or owned by public authorities from 1 January 2027. These targets will be achieved thanks to the concept of nearly zero-energy buildings (nZEB) [3,10,11,12,13,14]. nZEB buildings are characterized by high energy efficiency, with very low demand for energy from renewable energy sources [15,16,17,18,19,20,21,22]. Emission reduction is required throughout the life cycle of buildings, during construction, operation, and maintenance. Currently, more and more nZEB buildings are being built in EU Member States [23,24,25]. However, the gap in the energy efficiency of new buildings from the beginning of operation [8,26,27,28], resulting from climate change [23,29,30], makes it necessary to improve the energy efficiency of existing nZEB buildings as well. This energy efficiency gap is due to the fact that throughout Europe, most buildings were constructed before the introduction of thermal standards and energy efficiency regulations [31,32,33]. Therefore, in order to reduce energy consumption and greenhouse gas emissions, it is necessary not only to construct nZEB buildings but also to retrofit existing buildings. The key element to achieving acceptable thermal comfort inside buildings is their envelope [34].

Therefore, the challenge for the construction industry is to meet the growing demand for nZEB buildings and the circular economy [15,35,36]. Manufacturers of building materials are implementing innovative solutions that have the potential to enable the widespread construction of nZEB buildings. However, they encounter legal barriers related to the lack of standardization procedures enabling the rapid placement of innovative construction products on the market [37,38,39].

The Construction Product Regulation (CPR) evaluation [40] and feedback from Member States and stakeholders have clearly identified barriers to the functioning of the single market for construction products and thus the failure to achieve CPR objectives [37]. The conditions for circulating construction products on the European market will be amended. On 10 April 2024, the European Parliament adopted a legislative resolution at the first reading on the proposal for regulating the European Parliament and the Council, laying down harmonized conditions for the marketing of construction products [41]. However, it should be noted that the amended CPR [41] will not enter into force until a decade from now. The key changes to the CPR [41] include the following:Ensuring the smooth functioning of the single market and the free movement of construction products;Increasing the role of the sustainable use of natural resources in the life cycle of construction products;Enabling the construction sector to achieve climate and sustainability objectives and participate in the digital transformation of the economy;Ensuring competitiveness and eliminating barriers to harmonized assessment documents.

Project Horizon 2020: Measuring Envelope Products and Systems Contributing to the Next Generation of Healthy, Nearly Zero-Energy Buildings (MEZeroE), implemented in January 2021, aims to create an ecosystem combining the infrastructure resources and knowledge of academic and research centers with innovative solutions proposed by the industry in the form of a multi-sided virtual market open to the exchange of knowledge and experience between interested parties from the construction industry. The ecosystem will provide ready-made services in the fields of modeling, testing, and monitoring nZEB Enabler Envelope Technology Solutions (nEES) while creating a comprehensive knowledge management and training environment [42].

The project aims to consider the trends of Industry 4.0 in terms of fast decision making and customer orientation open to innovation, with particular emphasis on manufacturers of carbon-neutral construction products and healthy indoor environments [42].

The project tests standard, modified, and non-standard products using state-of-the-art equipment. Support for manufacturers in certifying the product and introducing it to the market is also provided. Standard tests have been extended to tests in live laboratories located throughout Europe, in which a new parameter of a non-engineering variable will be introduced, expressed through user opinions resulting from the comfort of use [42].

MEZeroE will be accessible online “https://mezeroe-platform.eu/ (accessed on 1 September 2024)” as a multi-sided virtual marketplace with a single entry point, including nine Pilot Measurement & Verification Lines (PM&VL) and three open innovation services (OIS) covering training, business model development, systematic intellectual property, and knowledge management. MEZeroE will accelerate the introduction of prototypes to the market as certified construction products. The final objective of the project is to develop information paths for the CE marking of different groups of construction products used in the building envelope. The paths will include procedures in accordance with Regulation 305/2011 (CPR) [40] or its amendment [41] and ways to quickly standardize and introduce innovative products to the market [42].

This article presents one of the research procedures developed and tested by Pilot Measurement & Verification Lines (PM&VL7) as part of the MEZeroE project for the polyurethane flexible connectors (PUFCs) offered by the MEZeroE project partner Flex&Robust. This procedure considers the applicable legal requirements regarding CE marking and indicates the certification path for this product type.

## 2. Legal Requirements for CE Marking

The path to the CE marking of construction products is specified in the still-applicable Regulation (EC) No. 305/2011 of the European Parliament and the Council of 9 March 2011, Construction Product Regulation (CPR) [40]. This regulation also specifies harmonized conditions for placing construction products on the market or making them available by establishing harmonized rules for expressing the performance of construction products in relation to their essential characteristics.

In accordance with the applicable law, this regulation requires implementation in European Union countries. In Poland, it was introduced by the Act of 16 April 2004 on construction products [43], which specifies the rules for placing construction products on the market. It approximates the implementing acts and laws of all EU Member States concerning construction products. According to Art. 5, paragraph 1 of this Act [43], a construction product that has been covered by a harmonized technical specification, i.e., a harmonized standard (hEN) or a European Technical Assessment (ETA), may be placed on the market or made available on the internal market only and exclusively in accordance with the provisions of the CPR [40].

The procedure for CE marking and then the placing of a construction product on the market in accordance with the CPR is conducted according to the scheme in Figure 1. According to the scheme, the European system [40] allows for the placement of a construction product on the market based on a harmonized technical specification, which is an hEN or a European Assessment Document (EAD).

In the case of the hEN (“harmonised standard” means a standard adopted by one of the European standardisation bodies listed in Annex I of Directive 98/34/EC on the basis of a request issued by the Commission in accordance with Article 6 of that Directive) for a construction product, CE marking and the placing of innovative products on the European market has a procedure established in Annex ZA of the hEN. For a construction product not covered or not fully covered by the hEN, an alternative harmonized technical specification, constituting the basis for CE marking, is the European Technical Assessment (ETA). ETAs are issued by Technical Assessment Bodies (TABs) based on EADs developed by the European Organisation for Technical Assessment (EOTA) and cited by the Commission in the Official Journal of the European Union [40]. An ETA entails a documented Assessment and Verification of Constancy of Performance (AVCP) of a construction product in relation to its essential characteristics in accordance with the relevant EAD. After conducting the AVCP, the manufacturer issues a Declaration of Performance (DoP) and marks the product with the CE mark on its basis.

## 3. Characterization of the Flex&Robust Polyurethane Flexible Connectors

Components of envelopes offered by Flex&Robust are structural and non-structural bonding connectors, constructing joints between various envelope components and between envelopes and structural elements.

The Flex&Robust line of products is dedicated to the innovative structural and non-structural bonding of elements constructing civil engineering structures made of various materials (concrete, masonry, wood, and metal). The specially designed products can carry static, dynamic, and cyclic loads and simultaneously transfer high deformations (Figure 2). They resist elevated temperatures and reduce stress concentrations by redistributing them over a large bonding area. Flex&Robust products can be used in seismic areas and strong wind areas as solutions that are resistant to repeatable loads and thus do not need to be replaced like other connectors after catastrophic events. They protect connected envelope components against damage.

A wide range of Flex&Robust products’ innovative properties can be adjusted to various requirements using special components. The Flex&Robust line of products can construct polyurethane flexible joints (PUFJs) and fiber-reinforced polyurethanes (FRPUs) that manifest a significant load and deformation capacity, fulfilling vibroacoustic and thermal comfort requirements, as well as manifesting durability, waterproofing, and non-conductive electric properties.

PUFJs reduce stress concentrations and distribute them across a larger surface, which allows large loads to be transferred. The innovative PUFJ is an effective structural connection in timber construction, an alternative to using dowel-type connections. Polyurethane construction PUFJs are stable at elevated temperatures up to 200 °C; thus, it is expected that PUFJs will work stably in fire conditions when they are protected by a timber layer.

### 3.1. Flex&Robust Layer (PUFJ)

One possible PUFC can be prefabricated or constructed on site: the flexible layer (PUFJ), which connects structural and non-structural envelop components. The Flex&Robust layer can be adjusted to any bonding surface shape and thickness and used in designing mechanical properties. As an example, a flexible layer (PUFJ) connecting a structural element (RC frame) and a non-structural envelope panel (infill masonry wall) is presented in Figure 3.

### 3.2. Flex&Robust Composite (FRPU)

Another possible PUFC is prefabricated or constructed on site: composite material (FRPU), which also connects structural and non-structural envelop components. The Flex&Robust composite can be adjusted to any bonding surface shape and thickness as well as used in mechanical property design. As before, a flexible composite (FRPU) connecting a structural element (RC frame) and a non-structural envelope panel (infill masonry wall) is presented in Figure 4.

### 3.3. Flex&Robust Injection (PUFJ)

The last possible PUFC solution is liquid injectable material with a fast curing time, which is suitable for filling gaps and bonding structural and non-structural envelop components, as well as for repairing damaged elements.

The Flex&Robust injection can be adjusted to any bonding surface shape and thickness (cracks/spaces) and used in mechanical property design. A flexible injection (PUFJ) connecting a structural element (RC frame) and a non-structural envelope panel (infill masonry wall) is presented in Figure 5.

## 4. PM&VL7 Test Line

The PM&VL7 research line Mechanical and durability tests of connectors and their impact on vibroacoustic, thermal and microclimate comfort was established at the Cracow University of Technology (CUT) as part of the MEZeroE project. An interdisciplinary team of 30 researchers proposed 69 types of varied tests offered to industrial partners within four sub-lines:(7.1) Mechanical;(7.2) Durability;(7.3) Vibroacoustic;(7.4) Thermal and microclimatic comfort.

Mechanical, vibroacoustic, and thermal tests were conducted as independent tests or constituted diagnostic features before and after aging. Within sub-line 7.2, durability tests cover aging and diagnostics before, after, and during aging, including microscopic observations and chemical tests (Figure 6), in addition to the tests mentioned above.

The main objective of PM&VL7 is the development of specific testing procedures for the evaluation of connector safety, with connectors considered as joints between envelope components and connections between envelope panels and structural elements. The proposed pilot line allows users to conduct typical tests on connections and examine their safety in use, with reference to vibroacoustic and thermo-humidity comfort (defined in European standards and not defined in the pilot line), e.g., for innovative solutions not covered by standards. As an added value, PM&VL7 includes additional tests characterizing the impact of environmental, chemical, biological, and mechanical factors (durability) on the reduction in safety and use of connections and the loss of vibroacoustic and thermo-humidity parameters of nZEB elements due to deteriorating factors.

## 5. Path of CE Marking for Polyurethane Flexible Connectors

An analysis of the certification requirements [40] enabling the determination of the CE marking path was conducted before defining the scope of research carried out by the PM&VL7 research line as part of the MEZeroE project.

### 5.1. Product Characteristics and Intended Use

The product type was determined: flexible polyurethane Flex&Robust connectors made of nine types of polyurethane two-component mass. The products are grouped in Table 1 depending on the type of polyurethane used.

The intended use was indicated as a repair system—the connection and strengthening of wooden, masonry, reinforced concrete, and stone structures exposed to dynamic loads, including seismic and strong wind loads.

The products are covered by two CPR [40] groups:25—construction adhesives;32—products for sealing joints.

### 5.2. Analysis of Harmonized Technical Specification

An analysis of the existing hENs and EDOs was conducted. No harmonized technical specification, neither hEN nor EDO, was found for the indicated construction products. Therefore, an analysis of hENs and EDOs for similar products was carried out, and, due to the indicated intended structural use, the AVCP systems presented in Table 2 were indicated as applicable.

Due to the lack of an existing reference document, the manufacturer is required to apply for an ETA.

### 5.3. AVCP

The AVCP system(s) of polyurethane polymer connectors is indicated in Table 3 under system 2+ without fire resistance and in Table 4 under system 3 for reaction to fire.

## 6. PM&VL7 Test Set for Polymer Flexible Connectors

The tests planned by PM&VL7 are aimed at determining the safety of connections in building partitions, treated as connections between partition elements and connections between partition plates and structural elements, along with checking the vibroacoustic and thermo-humidity comfort provided by polyurethane flexible joints. In addition, planned durability tests, together with structural, mechanical, and thermal diagnostics, aim to determine the impact of using connections on the safety and the loss of parameters of nZEB elements due to environmental impact.

Considering the possibilities and needs of both the MEZeroE project and the PM&VL7 research line, individual types of products were selected for testing, which are presented in Table 5. The most important tests were qualified for testing, the results of which will determine the continuation of the next product types. The sets of tests carried out within the individual sub-lines are presented in the following subsections.

### 6.1. Mechanical Tests

The dynamic stiffness/modulus of a composite according to EN 12697-26 [44] with modifications.The breaking force of the glass fiber grid according to ISO 3341 [45]—a diagnostic feature of durability due to XE aging.The matrix tensile strength and modulus of elasticity of polyurethane PS according to EN ISO 527-2 [46] and ISO 37 [47]—a diagnostic feature of durability due to XE aging.The composite tensile strength and modulus of elasticity according to ISO 527-4 [46]—the warp direction of polyurethane PS reinforced with a glass fiber grid, which is a diagnostic feature of durability due to XE aging and seawater aging.The initial shear strength includes two types of injection according to EN 1052-3 [48]—polyurethane PM and PST.The tensile strength and modulus of elasticity of two types of adhesive materials (polyurethanes PSTF-S and PSTF-W) used in layer products according to EN ISO 527-2 [46] and ISO 37 [47]—a diagnostic feature of durability due to XE ageing.

### 6.2. Durability Tests

The resistance to artificial aging by exposure to sunlight Xe composites, injection (polyurethane PM and PST), and two types of adhesive materials (polyurethanes PSTF-S and PSTF-W) used in layer products in accordance with the requirements of EN ISO 4892-2 [49]. Diagnostics of durability due to aging included features not defined in the standard. The tests before and after aging included mechanical features described in Section 6.1 and the following:
Observation under an optical microscope (OM);Observation under a scanning microscope (SEM);FTIR analyses.Resistance to artificial aging was determined using exposure to seawater according to an innovative procedure. Diagnostics of durability due to aging included features not defined in the standard. The tests before and after aging included mechanical features described in Section 6.1.

### 6.3. Acoustic Tests

The critical damping ratio [%] according to a modified EN 29052-1 procedure [50].The airborne and impact sound insulation measured for the prepared solution according to the innovative procedure.The direction-averaged junction velocity level difference for a connector or connection model according to an innovative procedure based on EN ISO 12354-1 [51] and EN ISO 12354-2 [52].The dynamic stiffness (MN/m^3^) according to EN 29052-1 [50].

### 6.4. Thermal Simulations

Water vapor diffusion (interstitial water vapor condensation risk and intensity) according to EN ISO 13788 [53].Internal surface temperature according to EN ISO 13788 [53].Linear thermal transmittance according to EN ISO 10211 [54].

## 7. Summary

This article presents the certification path for innovative flexible polyurethane connectors tested within the MEZeroE project. Developed in the PM&VL7 research line at the Cracow University of Technology, this certification path results from cooperation with the product manufacturer FlexAndRobust Systems Ltd. It implements the key objective of the MEZeroE project—supporting the manufacturer in the field of CE marking—which accelerates the placement of the product on the market.

Polyurethane flexible connectors are designed to repair and strengthen structures, such as those made of wood, masonry, reinforced concrete, and stone, that are exposed to dynamic loads like seismic activity and strong winds. These connectors provide a durable and flexible solution for connecting and reinforcing structural components, helping to mitigate damage and improve overall structural integrity. Due to the lack of an existing reference document, the manufacturer was required to apply for European Technical Assessment (ETA).

This article also indicates the Assessment and Verification of Constancy of Performance (AVCP) systems for polyurethane flexible connectors, including system 2+ without fire resistance and system 3 regarding the reaction to fire. The scope of the research in four key areas—mechanical, strength, thermal, and acoustic— was defined using the PM&VL7 research line.

The example described in the article also highlights the potential of the service platform launched as part of the MEZeroE project at “https://mezeroe-platform.eu/ (accessed on 1 September 2024)”, which aims to overcome legal barriers related to the lack of standardization procedures enabling the rapid placement of innovative construction products on the market.

## Figures and Tables

**Figure 1 materials-17-05503-f001:**
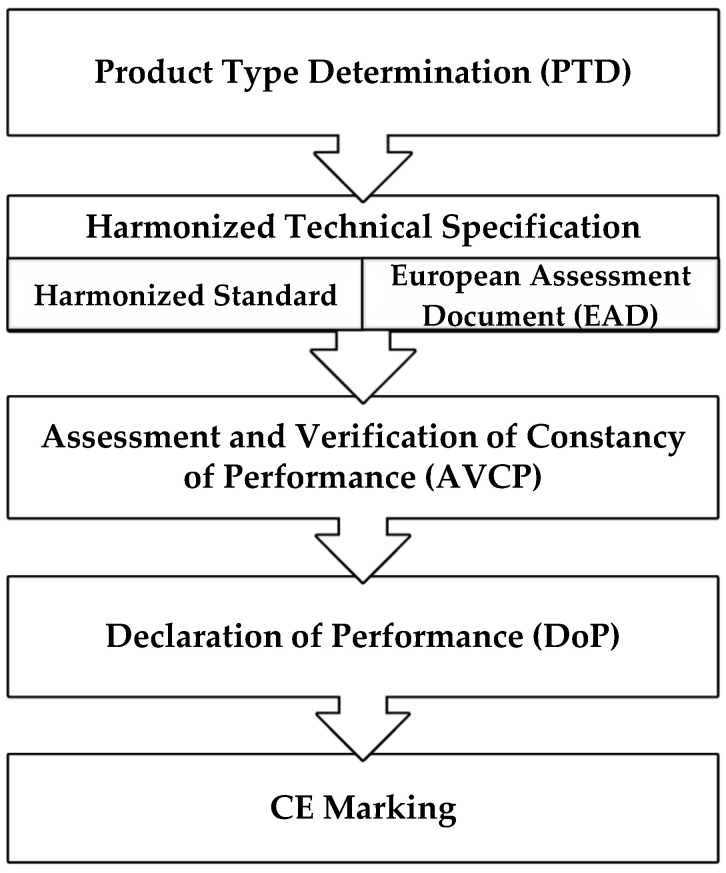
A diagram of the CE marking path [40].

**Figure 2 materials-17-05503-f002:**
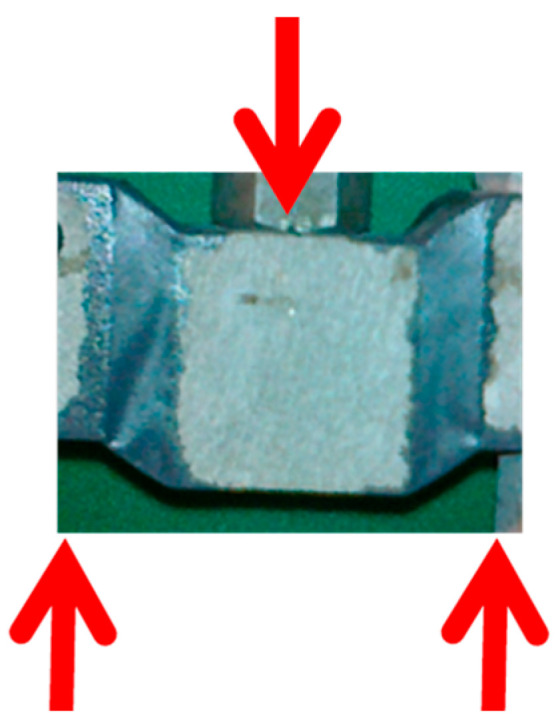
Visualization of the polyurethane flexible joint transferring large deformations under load marked with arrows.

**Figure 3 materials-17-05503-f003:**
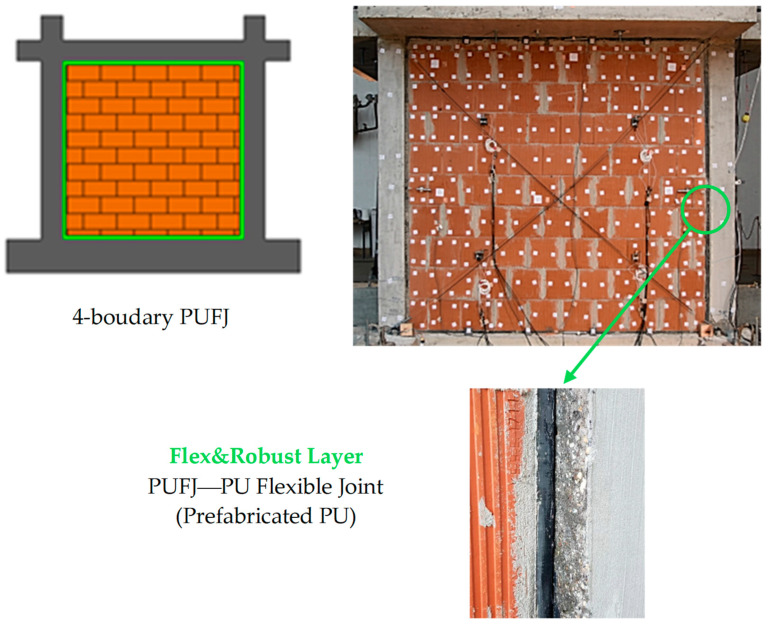
A visualization of the prefabricated Flex&Robust layer (PUFJ).

**Figure 4 materials-17-05503-f004:**
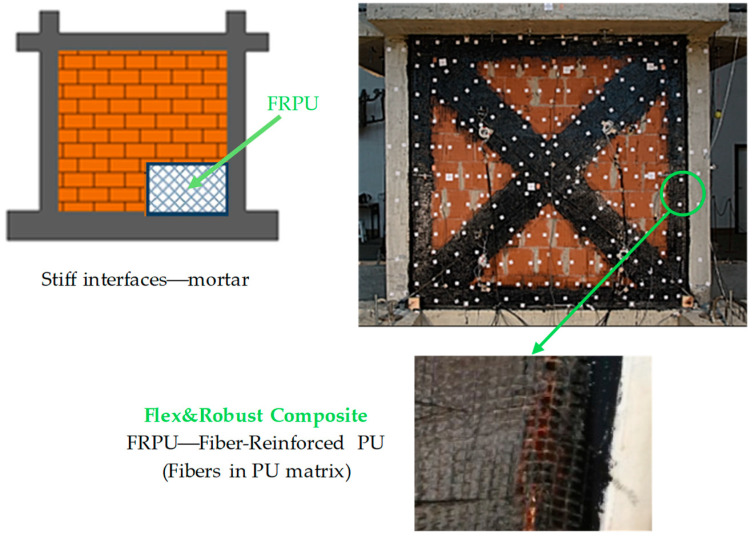
**A** visualization of the Flex&Robust composite (FRPU).

**Figure 5 materials-17-05503-f005:**
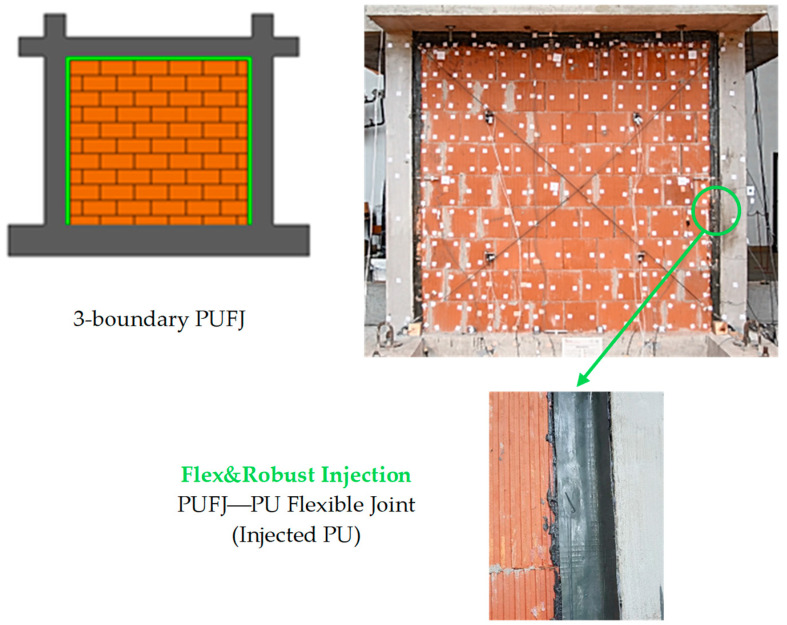
A visualization of Flex&Robust injection (PUFJ).

**Figure 6 materials-17-05503-f006:**
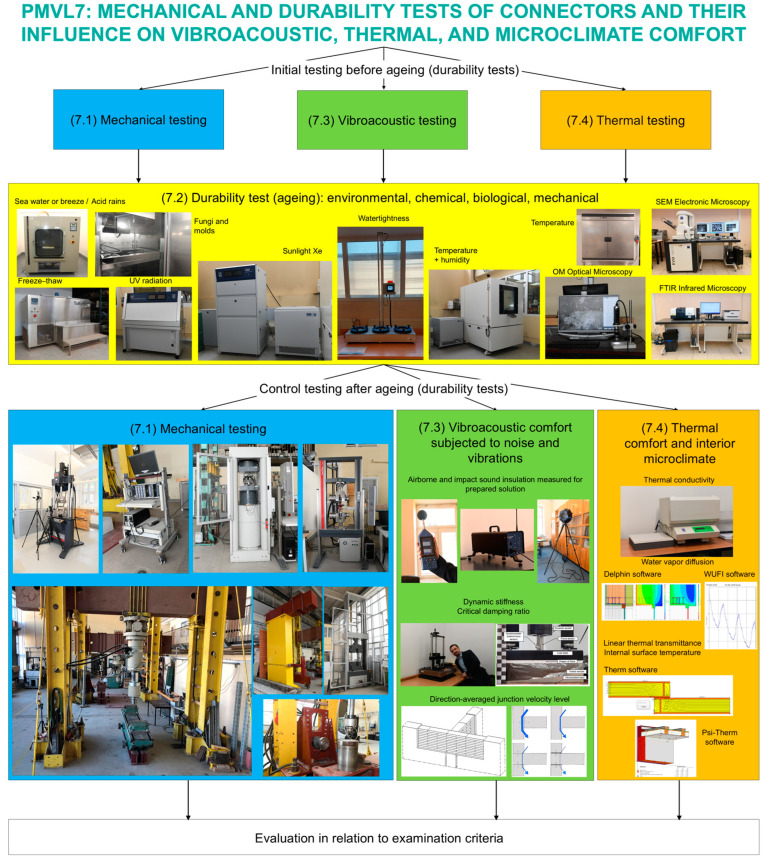
A schematic description of PM&VL7.

**Table 1 materials-17-05503-t001:** Flex&Robust products.

Product—Polyurethane Flexible Connector	Type of Materials
Flex&Robust Composite	Polyurethane PS reinforced with glass fiber grid
Flex&Robust Layer	Polyurethane PSTF-S Polyurethane PSTF-W Polyurethane PM
Flex&Robust Injection	Polyurethane PM Polyurethane PST

**Table 2 materials-17-05503-t002:** Systems of AVCP [40].

Product(s)	Intended Use(s)	AVCP System(s) ^1^
Polymer Flexible Connectors: Flex&Robust Injections; Flex&Robust Composites: polyurethanes reinforced with glass fiber mesh; Flex&Robust Layers.	Any	2+
Polymer Flexible Connectors: Flex&Robust Injections; Flex&Robust Layers.	For uses subject to regulations on reaction to fire	3

^1^ The systems of Assessment and Verification of Constancy of Performance of Construction Products on a scale of 1, 1+, 2+, 3, and 4 are specified in Annex V of the CPR [40]. Individual AVCPs distinguish the scope of tasks of the manufacturer and notified bodies: product certification, production control or testing laboratories.

**Table 3 materials-17-05503-t003:** Assignment of AVCP tasks under system 2+.

Tasks for the manufacturer	Factory production control (FPC) Determination of the product type on the basis of type testing (including sampling), type calculations, tabulated values, or descriptive documentation of the product Further testing of samples taken at the factory according to the prescribed test plan
Tasks for the notified factory production control certification body	Initial inspection of the manufacturing plant and of FPC Continuation of surveillance, assessment, and evaluation of FPC

**Table 4 materials-17-05503-t004:** Assignment of AVCP tasks under system 3 for reaction to fire.

Tasks for the manufacturer	Factory production control (FPC)
Tasks for the notified laboratory	Determination of the product type on the basis of type testing (including sampling), type calculations, tabulated values, or descriptive documentation of the product

**Table 5 materials-17-05503-t005:** Flex&Robust product used for the tests.

Product—Polymer Flexible Connector	Type of Materials
Flex&Robust Composite	Polyurethane PS matrix reinforced by glass fiber grid
Flex&Robust Layer	Polyurethane PSTF-S Polyurethane PSTF-W Polyurethane PM
Flex&Robust Injection	Polyurethane PM Polyurethane PST

## Data Availability

The original contributions presented in the study are included in the article, further inquiries can be directed to the corresponding author.

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
