# Peer review of "A Roadmap for the Certification of Polyurethane Flexible Connectors Used as Envelope Products in the Next Generation of Healthy, Nearly Zero-Energy Buildings"

_materials, 2024, doi:10.3390/ma17225503_

Round 1
Reviewer 1 Report
Comments and Suggestions for Authors
It is a very good paper in al its parts and could be published without any comment, well presented and scientifically complete.
In chapter 3.3 I don't see the caption for the image, the first one, maybe you should find the way to connect it with the ones in the next page.
Maybe it would be more functional to use the IMRAD form, considering the quality of the paper as the last part written as "Summary" looks not at the same level as the rest of the paper.
Those are just minor remarks, the paper is a very good scientific work.
Author Response
Cracow, Poland 18-10-2024
Dear Reviewer,
Many thanks for your comments. Below I have replied to the comments contained in the review.
Comments 1: It is a very good paper in all its parts and could be published without any comment, well presented and scientifically complete.
Response 1: Thank you.
Comments 2: In chapter 3.3 I don't see the caption for the image, the first one, maybe you should find the way to connect it with the ones in the next page.
Response 2: Thank you for pointing this out. The image in chapter 3.3 has been merged.
Comments 3: Maybe it would be more functional to use the IMRAD form, considering the quality of the paper as the last part written as "Summary" looks not at the same level as the rest of the paper.
Response 3: Thank you for pointing this out. In accordance with the recommendations of Reviewer 3, the summary has been modified. I leave it to the Publisher to decide whether to publish the article in IMRAD form or leave the summary.
Comments 4: Those are just minor remarks, the paper is a very good scientific work.
Response 4: Thank you.
The article was corrected in accordance with the recommendations of Reviewers.
The article has been sent to the MDPI language proofreader.
I hope that the revised manuscript now fully meets the criteria for publication in Materials. Thank you very much for your efforts concerning my manuscript.
Yours faithfully,
Aneta Nowak-Michta, Arkadiusz Kwiecień, Jagoda Michta
Reviewer 2 Report
Comments and Suggestions for Authors
Review manuscript ID: materials-3274358
Title: Roadmap for certification of polyurethane flexible connectors used as envelope products in the next generation of healthy, nearly-zero energy buildings
Authors: Aneta Nowak-Michta*, Arkadiusz Kwiecień, Jagoda Michta
Journal: Materials
Type of manuscript: Article
Section: Polymeric Materials
Special Issue: Polymers, Processing and Sustainability
Assistant Editor: Ms. Hilaria Li
Publisher: MDPI, Basel, Switzerland
Date: 14 October 2024
This article outlines a research procedure developed by the Horizon 2020 project’s Pilot Measurement & Verification Lines (PM&VL7) for FLEX&ROBUST polyurethane flexible connectors to assist producers of innovative envelope products used in zero-energy buildings, particularly in certification, CE marking, and market placement. The following provides some remarks.
* When referring to Europe, what do you have in mind? Europe is quite diverse, and regulations from one country to another vary.
* What is CE?
* In Figure 2, what do the arrows indicate?
* What is the meaning of 2+ and 3 in Table 2? What would be the full scale?
* Some mathematical and physical aspects of the tests outlined in Section 6 could be mentioned briefly. Additionally, I wonder whether the insulation test could be included there. The rationale is that better quality insulation will prevent heat from escaping, especially during cold winter days, and thus save energy tremendously. Please refer to that documentary from BBC that compared the British and German windows, and the winner is obvious.
* Please adjust the margin in References No. [5, 6], etc.
* Some references are missing the page numbers. Consider following the format requirement from the journal/publisher.
* The abstract can be improved. Here is a suggested version:
Currently, in Europe, producers of building materials are implementing innovative solutions that provide a chance for the widespread construction of zero-emission and zero-energy buildings. However, they encounter legal barriers related to the lack of standardization procedures to enable the rapid placement of innovative construction products on the market. The European project Horizon 2020: Measuring Envelope Products and Systems Contributing to the Next Generation of Healthy Nearly Zero Energy Buildings (MEZeroE) aims to support producers of innovative envelope products used in zero-energy buildings, including in the field of certification, CE marking, and placing them relatively quickly on the market. The article presents one of the research procedures developed and tested by Pilot Measurement & Verification Lines (PM&VL7) as part of the MEZeroE project for FLEX&ROBUST polyurethane flexible connectors. This procedure takes into account applicable legal requirements regarding CE marking and also indicates a certification path for this type of product.
Grammar issue.
Author Response
Cracow, Poland 18-10-2024
Dear Reviewer,
Many thanks for your comments. Below I have replied to the comments contained in the review.
Comments 1: When referring to Europe, what do you have in mind? Europe is quite diverse, and regulations from one country to another vary.
Response 1: Thank you for pointing this out. In the article, line 11 clarifies that it concerns the European Economic Area (EEA)
Comments 2: What is CE?
Response 2: Thank you for pointing this out. In line 18 the explanation of the CE marking has been added.
Comments 3: In Figure 2, what do the arrows indicate?
Response 3: Thank you for pointing this out. In the description of Figure 2, ‘marked with arrows’ has been added.
Comments 4: What is the meaning of 2+ and 3 in Table 2? What would be the full scale?
Response 4: The AVCP (Systems of Assessment and Verification of Constancy of Performance of construction products - the abbreviation has been previously defined in the text of the article) specified in Table 2 are in accordance with CPR (REGULATION (EU) No 305/2011 OF THE EUROPEAN PARLIAMENT AND OF THE COUNCIL of 9 March 2011 laying down harmonised conditions for the marketing of construction products and repealing Council Directive 89/106/EEC. 2011: Official Journal of the European Union) [40]. A detailed description of the AVCP of all systems 1, 1+, 2+, 3 and 4 would take up a lot of space, therefore the authors request the Reviewer to agree not to publish the supplement of this scope, which readers can find in CPR [40].
Comments 5: Some mathematical and physical aspects of the tests outlined in Section 6 could be mentioned briefly. Additionally, I wonder whether the insulation test could be included there. The rationale is that better quality insulation will prevent heat from escaping, especially during cold winter days, and thus save energy tremendously. Please refer to that documentary from BBC that compared the British and German windows, and the winner is obvious.
Response 5: The insulation test was included in point 6.4, because the simulations mentioned there require material data on the thermal conductivity coefficient or vapor permeability. However, due to the fact that the article covers only the certification path and not its full description, the tests have not been fully described. The authors request the Reviewer to include this explanation.
Comments 6: Please adjust the margin in References No. [5, 6], etc.
Response 6: Thank you for pointing this out. The margins in the bibliography have been adjusted.
Comments 7: Some references are missing the page numbers. Consider following the format requirement from the journal/publisher.
Response 7: Thank you for pointing this out. The authors are aware of the lack of page numbering on some pages. The article uses an automatic literature system that will restore the previous state when editing.
Comments 8: The abstract can be improved. Here is a suggested version:
Currently, in Europe, producers of building materials are implementing innovative solutions that provide a chance for the widespread construction of zero-emission and zero-energy buildings. However, they encounter legal barriers related to the lack of standardization procedures to enable the rapid placement of innovative construction products on the market. The European project Horizon 2020: Measuring Envelope Products and Systems Contributing to the Next Generation of Healthy Nearly Zero Energy Buildings (MEZeroE) aims to support producers of innovative envelope products used in zero-energy buildings, including in the field of certification, CE marking, and placing them relatively quickly on the market. The article presents one of the research procedures developed and tested by Pilot Measurement & Verification Lines (PM&VL7) as part of the MEZeroE project for FLEX&ROBUST polyurethane flexible connectors. This procedure takes into account applicable legal requirements regarding CE marking and also indicates a certification path for this type of product.
Response 8: Thank you for pointing this out. The recommended corrections were made and the article was also subjected to linguistic correction.
The article was corrected in accordance with the recommendations of Reviewers.
The article has been sent to the MDPI language proofreader.
I hope that the revised manuscript now fully meets the criteria for publication in Materials. Thank you very much for your efforts concerning my manuscript.
Yours faithfully,
Aneta Nowak-Michta, Arkadiusz Kwiecień, Jagoda Michta
Reviewer 3 Report
Comments and Suggestions for Authors
Comment:
The article presents one of the research procedures developed and tested by Pilot Measurement & Verification Lines (PM&VL7) as part of the MEZeroE project for FLEX&ROBUST polyurethane flexible connectors. This procedure takes into account applicable legal requirements regarding CE marking and also indicates a certification path for this type of products.
From my side, this is an interesting work. However, it still should be well revised before it can be considered.
1) Please do not divide the abstract into two paragraphs.
2) Figure 3. Visualization of the prefabricated Flex&Robust Layer (PUFJ). & Figure 5. Visualization of the Flex&Robust Injection (PUFJ).
These pictures are too vague. Please modify them carefully.
3) The conclusion of this article is missing. And the “5. Summary” This section is too loose to see the innovation and academic contribution of your work. Please well revise it.
4) The references are not standardized, especially with many unauthoritative reference sources. For instance ref.[4] [5][6][42].
Comments on the Quality of English LanguageIt should be well revised before it can be considered.
Author Response
Cracow, Poland 18-10-2024
Dear Reviewer,
Many thanks for your comments. Below I have replied to the comments contained in the review.
Comments 1: From my side, this is an interesting work.
Response 1: Thank you.
Comments 2: Please do not divide the abstract into two paragraphs.
Response 2: Thank you for pointing this out. The abstract division has been removed.
Comments 3: Figure 3. Visualization of the prefabricated Flex&Robust Layer (PUFJ). & Figure 5. Visualization of the Flex&Robust Injection (PUFJ).
Response 3: Thank you for pointing this out. Figures 3-5 have been corrected.
Comments 4: The conclusion of this article is missing. And the “5. Summary” This section is too loose to see the innovation and academic contribution of your work. Please well revise it.
Response 4: Thank you for pointing this out. The summary has been corrected, but one of the reviewers recommended leaving the article in the IMRAD format. Therefore, the authors leave it to the Editor to decide whether to leave the corrected summary or remove it.
Comments 5: The references are not standardized, especially with many unauthoritative reference sources. For instance ref.[4] [5][6][42].
Response 5: Thank you for pointing this out. The bibliographic descriptions have been corrected, however, they were entered automatically and the authors are concerned that they will revert to their previous state when editing.
The article was corrected in accordance with the recommendations of Reviewers.
The article has been sent to the MDPI language proofreader.
I hope that the revised manuscript now fully meets the criteria for publication in Materials. Thank you very much for your efforts concerning my manuscript.
Yours faithfully,
Aneta Nowak-Michta, Arkadiusz Kwiecień, Jagoda Michta
Round 2
Reviewer 2 Report
Comments and Suggestions for Authors
Second review manuscript ID: materials-3274358
Title: Roadmap for certification of polyurethane flexible connectors used as envelope products in the next generation of healthy, nearly-zero energy buildings
Authors: Aneta Nowak-Michta*, Arkadiusz Kwiecień, Jagoda Michta
Journal: Materials
Type of manuscript: Article
Section: Polymeric Materials
Special Issue: Polymers, Processing and Sustainability
Assistant Editor: Ms. Hilaria Li
Publisher: MDPI, Basel, Switzerland
Date: 23 October 2024
Thank you for revising your manuscript.
What I meant with the full scale is the complete range, not the full description of the references, which indeed readers can find out by themselves if they are interested.
The second stage of the diagram in Figure 1, what does it mean, or what are the differences between, the two standards?
The text labels in Figure 3 have low resolution. Please improve the resolution.
The same remark for other figures applies.
Clarity: submit an application --> apply
This sentence can be improved:
"The purpose of polyurethane flexible connectors was as a repair system for connecting and strengthening wooden, masonry, reinforced concrete, and stone structures exposed to dynamic loads, including seismic and strong wind loads."
Here is a possible improvement:
"Polyurethane flexible connectors are designed to repair and strengthen structures, such as those made of wood, masonry, reinforced concrete, and stone, that are exposed to dynamic loads like seismic activity and strong winds. These connectors provide a durable and flexible solution for connecting and reinforcing structural components, helping to mitigate damage and improve overall structural integrity."
See the comment.
Author Response
Dear Reviewer,
Many thanks for your comments. Below I have replied to the comments contained in the
review.
Comments 1: What I meant with the full scale is the complete range, not the full description
of the references, which indeed readers can find out by themselves if they are interested.
Response 1: Thank you for pointing this out. Table 2 provides an explanation of AVCP
systems. In addition, the full scale of the systems is given in the footnote: ‘The systems of
Assessment and Verification of Constancy of Performance of Construction Products on a scale
of 1, 1+, 2+, 3 and 4 are specified in Annex V of the CPR [40]. Individual AVCPs distinguish
the scope of tasks of the manufacturer and notified bodies: product certification, production
control or testing laboratories’.
Comments 2: The second stage of the diagram in Figure 1, what does it mean, or what are the
differences between, the two standards?
Response 2: Thank you for pointing this out. The differences between the construction product
marking paths shown in Figure 1 have been completed, and they are described in lines 125-148.
Comments 3: The text labels in Figure 3 have low resolution. Please improve the resolution.
Response 3: Thank you for pointing this out. The drawings have been improved,
unfortunately it was difficult to see on a small monitor, for which I apologize.
Comments 4: The same remark for other figures applies.
Response 4: Thank you for pointing this out. The drawings have been improved, unfortunately
it was difficult to see on a small monitor, for which I apologize.
Comments 5: Clarity: submit an application --> apply
Response 5: Thank you for pointing this out. In line 334 the text has been changed according
to the Reviewer's recommendations.
Comments 6: This sentence can be improved:
"The purpose of polyurethane flexible connectors was as a repair system for connecting and
strengthening wooden, masonry, reinforced concrete, and stone structures exposed to dynamic
loads, including seismic and strong wind loads."
Here is a possible improvement:
"Polyurethane flexible connectors are designed to repair and strengthen structures, such as those
made of wood, masonry, reinforced concrete, and stone, that are exposed to dynamic loads like
seismic activity and strong winds. These connectors provide a durable and flexible solution for
connecting and reinforcing structural components, helping to mitigate damage and improve
overall structural integrity."
Response 6: Thank you for pointing this out. Lines 329-333 of the text have been changed
according to the Reviewer's recommendations.
The article has been corrected in accordance with the Reviewer's recommendations.
We hope that the revised manuscript now fully meets the criteria for publication in Materials.
Thank you very much for your efforts concerning our manuscript.
Yours faithfully,
Aneta Nowak-Michta, Arkadiusz Kwiecień, Jagoda Michta

Reviewer 3 Report
Comments and Suggestions for Authors
After revision, the manuscript has been improved significantly. now, it can be considered.
Author Response
Cracow, Poland 28-10-2024
Dear Reviewer,
Thank you very much for your efforts concerning our manuscript.
Yours faithfully,
Aneta Nowak-Michta, Arkadiusz Kwiecień, Jagoda Michta
Round 3
Reviewer 2 Report
Comments and Suggestions for Authors
Third review manuscript ID: materials-3274358
Title: Roadmap for certification of polyurethane flexible connectors used as envelope products in the next generation of healthy, nearly-zero energy buildings
Authors: Aneta Nowak-Michta*, Arkadiusz Kwiecień, Jagoda Michta
Journal: Materials
Type of manuscript: Article
Section: Polymeric Materials
Special Issue: Polymers, Processing and Sustainability
Assistant Editor: Ms. Hilaria Li
Publisher: MDPI, Basel, Switzerland
Date: 7 November 2024
Thank you again for meticulously revising your manuscript.
Consider using double quotations instead of single quotations.
Are the parentheses on Line 139 intentional?
Otherwise, the manuscript looks good and can be recommended for publication.
Best wishes and congratulations!
.
Author Response
Cracow, Poland 07-11-2024
Dear Reviewer,
Many thanks for your comments. Below I have replied to the comments contained in the review.
Comments 1: Consider using double quotations instead of single quotations.
Response 1: At the Reviewer's suggestion, the single quote on line 135 has been replaced with double quotes.
Comments 2: Are the parentheses on Line 139 intentional?
Response 2: Thanks for pointing that out. The parentheses on line 139 have been removed.
The article has been corrected in accordance with the Reviewer's recommendations.
We hope that the revised manuscript now fully meets the criteria for publication in Materials. Thank you very much for your efforts concerning our manuscript.
Yours faithfully,
Aneta Nowak-Michta, Arkadiusz Kwiecień, Jagoda Michta